# Contribution of paranasal sinus, chest, and abdomen/pelvis computed tomography in patients with febrile neutropenia

Charles Tran[1], Éric de Kerviler[2], Anne Bergeron [3], Emmanuel Raffoux [4], Aliénor Xhaard[5], Cédric de Bazelaire[2], Constance de Margerie-Mellon [2]*

1 Department of Medical Imaging, Hôpital National d'Instruction des Armées Bégin, Saint-Mandé, France, 2 Department of Radiology, Hôpital Saint-Louis, Assistance Publique-Hôpitaux de Paris, Paris, France and Université Paris Cité, Paris, France, 3 Department of Pneumology, Geneva University Hospitals, University of Geneva, Geneva, Switzerland, 4 Service Hématologie Adultes, Hôpital Saint-Louis, Assistance Publique-Hôpitaux de Paris, THEMA Saint-Louis Leukemia Institute, Université Paris Cité, Paris, France, 5 Service Hématologie-Greffe, Hôpital Saint-Louis, Assistance Publique-Hôpitaux de Paris, Université Paris Cité, Paris, France

* constance.de-margerie@aphp.fr

## Abstract

### Objective

Febrile neutropenia (FN) is a serious clinical event, associated with significant morbidity and mortality. Imaging has a central role in the identification of the fever cause. The study objectives were to assess abnormalities of potential infectious origin on paranasal sinus, chest and abdomen/pelvis CT scans performed during an episode of FN, in patients with or without specific clinical signs, and to evaluate their impact on the frequency of changes in treatment.

### Materials and methods

This retrospective study included 306 patients with FN from a single academic center between October 2018 and December 2019. Clinical and biological data, final clinical diagnosis for the FN episode, and change in treatment following CT scans were collected from medical records. CT images were reviewed for positive findings in each anatomical region.

### Results

Overall, 188 patients (61.4%) had at least one positive CT finding on paranasal sinus, chest and/or abdomen/pelvis regions, and 153 patients (50.0%) received a final clinical diagnosis of infection, based on microbiological and/or radiological findings. There were more chest and abdomen/pelvis CT positive findings in patients with specific clinical signs than in patients without (65.7% vs. 25.8%, and 59.0% vs. 22.7% respectively, p <0.001), with a higher frequency of therapeutic modifications in positive CTs. There were significantly more paranasal sinus CT abnormalities in symptomatic patients (33.3% vs. 8.5%, p = 0.03), but without any impact on treatment.

**Data Availability Statement:** According to the current French regulation, authorization to access patient data is given individually to the researcher

who has made the request, if this request is judged as acceptable, through the IRB. Therefore, this researcher cannot openly share the data, even anonymized. However, access to data is possible for other researchers on request (aphp-accesdonnees@aphp.fr).

**Funding:** The author(s) received no specific funding for this work.

**Competing interests:** The authors have declared that no competing interests exist

## Conclusion

These results support recommendations to perform systematic chest CT in patients with FN and may encourage the widespread use of abdomen/pelvis CT in this indication, even without symptoms. However, the actual impact of their use on the prognosis of the patients remains to be evaluated.

## Introduction

Febrile neutropenia (FN) is a critical yet common condition affecting onco-hematology patients [1]. It is defined as an oral temperature of >38.3˚C or two consecutive readings of >38.0˚C for 2 hours associated with an absolute neutrophil count (ANC) below 500/mm$^3$ or expected to fall below this threshold [2]. FN is associated with prolonged hospital stay, delay in the overall chemotherapy treatment plan, and mortality that can reach 40% in high-risk patients [2]. Immediate mortality is secondary to uncontrolled infections of bacterial, fungal or viral origins [3].

Patients with FN usually receive empirical broad-spectrum antibiotic therapy [2]. However, identifying the precise cause of fever is important to select the most adequate treatment, while limiting potential side effects of unnecessary drugs. It is a challenging task because signs and symptoms of inflammation are typically attenuated in the context of neutropenia. Moreover, non-infectious causes of fever such as certain medication use, blood transfusions, radiation, and neoplastic fever are not rare [4]. Infections are clinically documented in only 20%–30% of febrile episodes [5]. Along with extensive microbiological laboratory testing, imaging plays a central role, contributing to identify potential sources of infection, to approach the final appropriate diagnosis, and to guide therapeutic strategies. The contribution of chest computed tomography (CT) has already been investigated in FN for the early detection and management of invasive fungal infections such as aspergillosis [6–8]. However, the role of paranasal sinus and abdomen/pelvis CT, sometimes combined with chest CT in clinical routine to explore FN, has scarcely been studied.

Therefore, the main objective of our study was to identify abnormalities of potential infectious origin detected on paranasal sinus, chest and abdomen/pelvis CT scans performed in the setting of FN episode, in patients with or without specific clinical signs. The secondary objective was to assess the frequency of treatment changes following CT scan report.

## Materials and methods

### Study protocol and patient selection

This monocentric retrospective study was conducted after approval from the French National Ethics Committee on Medical Imaging Research (CRM-2107-182) that waived the need for informed consent in Saint-Louis Hospital (Assistance Publique-Hôpitaux de Paris), Paris, France. This hospital is a 500-bed academic center largely dedicated to treatment of cancers, including hematological malignancies and solid tumors. The study spanned over a 15-month period from October 1$^{st}$, 2018 to December 31$^{st}$, 2019. For the needs of the study, data were accessed from August 1$^{st}$, 2021 to October 31$^{st}$ 2021. Authors had access to data that could identify patients during the data collection step. Data were then deidentified for the data analysis step.

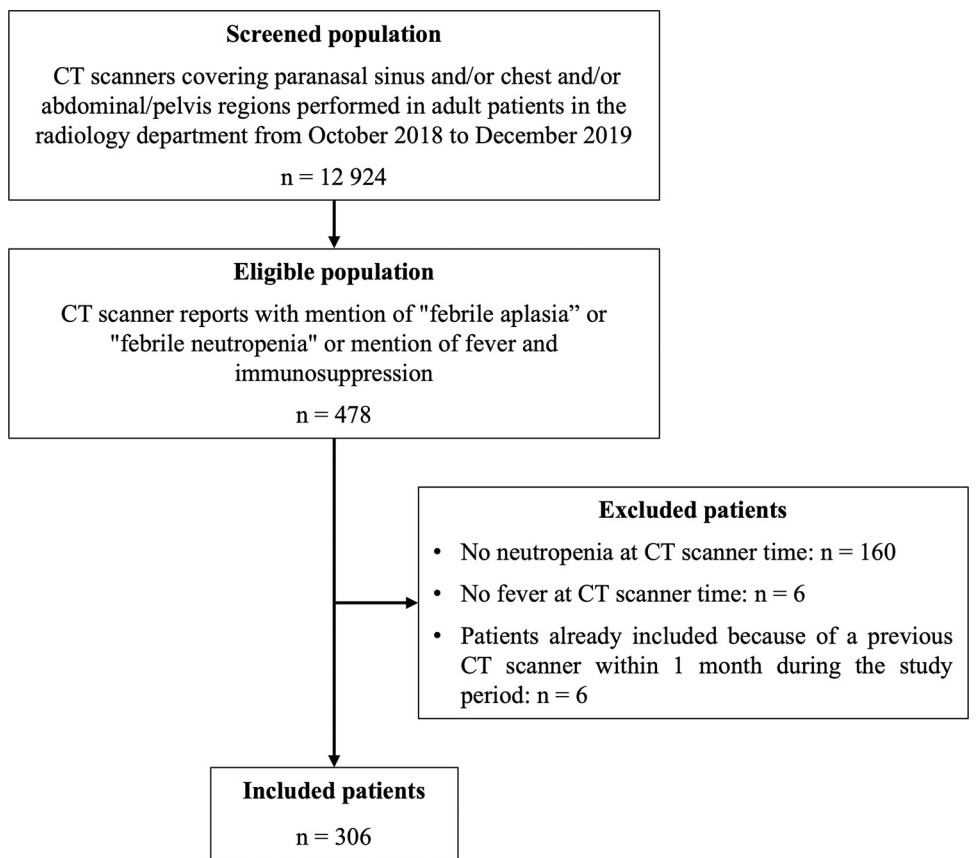

**Fig 1. Flow chart.** CT: Computed tomography.

The patient selection process consisted of reviewing the reports of all CT scans covering paranasal sinus and/or chest and/or abdomen/pelvis regions performed in adult patients in the radiology department during the study period. The keywords "febrile aplasia" and "febrile neutropenia" were searched, as well as mentions of a fever in patients undergoing chemotherapy, immunosuppressive therapies, followed-up for hematological diseases or solid cancer and/or who had undergone stem cell transplantation.

Once the eligible patients were identified by scanning the CT reports, medical files of each patient were studied, and the following inclusion criteria were applied (Fig 1): i) availability of clinical, biological, and imaging data; ii) history of hematological malignancy, solid cancer, immunosuppressive therapies, stem cell transplantation, or other cause of immunosuppression (aplastic anemia, inflammatory disease treatment with immunosuppressive drug, others), iii) neutropenia defined as ANC < 500/mm$^3$, iv) mention of fever (oral temperature of >38.3˚C or two consecutive readings of >38.0˚C) concomitant to neutropenia in the hospitalization reports. On the opposite, patients with missing clinical, biological and/or imaging data, no history of immunosuppression, and neutropenia without fever or fever without neutropenia were excluded. For patients who had undergone more than one CT within a month during the same episode of FN, only the first CT was considered.

Out of 478 eligible patients identified from the CT reports, 172 were excluded based on the above-mentioned criteria (Fig 1), and 306 patients were included.

## Data collection

**Clinical data.** Sex, age and body-mass index (BMI) were recorded for all patients. The primary conditions for which patients were being treated when FN occurred were obtained from the clinical records and categorized as follows

- Acute myeloid leukemia (AML) and acute lymphoblastic leukemia (ALL), not treated with HSCT nor CAR T-cell therapy

- Hematopoietic stem cell transplant (HSCT) recipients (autologous or allogeneic), whatever the underlying cause of transplantation

- Lymphoma, chronic lymphocytic leukemia (CLL), and multiple myeloma (MM), not treated with HSCT nor CAR T-cell therapy

- Myeloproliferative neoplasm (MPN) and myelodysplastic syndrome (MDS), not treated with HSCT

- Patients undergoing CAR T-cell therapy (for ALL or lymphoma)

- Solid cancer patients

- Others

The duration of neutropenia was recorded in days and categorized as short or prolonged using a 14-day cutoff [9]. The duration of FN was also recorded in days.

The presence or absence of clinical signs, categorized by system (nose/paranasal sinus, respiratory, digestive/urological), was documented from the clinical files and the CT orders. The final clinical diagnosis, as determined by physicians in the hospitalization departments and documented in the hospitalization report, was collected and categorized as infectious or non-infectious. Neutropenic enterocolitis, although being initially an inflammatory disease, was categorized as infectious because of the high risk of associated bacteremia and the need for broad-spectrum antibiotics [10]. Any treatment changes following CT scan results, including alterations in antibiotic, antifungal, antiviral or corticosteroid therapies, were recorded from the clinical files.

**Biological data.** White blood cell counts (WBC) and, if available, C-reactive protein (CRP) levels closest to the CT scan date, were recorded. The center where the study was performed does not report neutrophil counts when WBC $\leq$ 200/mm$^3$. Therefore, ANC were categorized in a binary way based on this threshold.

**Imaging data.** CT scan data included the examination date, the scanned anatomical regions scanned (paranasal sinus, chest, and abdomen/pelvis areas) and the use of contrast medium.

When abnormalities compatible with infection were noted in CT scan reports, CT images were reviewed jointly by a 5[th] year radiology resident and a board-certified radiologist with 8 years of experience, on a PACS workstation and categorized based on their main features. Paranasal sinus CT scans were assessed for sinusitis signs defined as gas-fluid levels, gas bubbles, or soft-tissue opacification of sinus with or without local invasion. Chest abnormalities were assessed for signs of bacterial, viral or fungal pneumonia, including nodules (that could be solid, ground glass, with halo sign, with cavitation), micronodules, parenchymal consolidations (with or without nodules or cavitation), ground glass opacities, and for pleural effusion. Abdomen/pelvis CT scans were examined for digestive wall thickening, hypo enhancements and nodules in solid organs, gallbladder thickening and distension, pyelic wall thickening, peritoneal effusions, and soft tissue collections. In each anatomical region, the above-described findings were defined as CT positive findings.

## Statistical analysis

Categorical data were presented as numbers and percentages. Continuous variables, all deviating from a normal distribution (confirmed by Shapiro-Wilk and Kolmogorov-Smirnov tests), were expressed as median with interquartile range (IQR). Chi2 tests, Fisher tests, Mann-Whitney tests, or Kruskal-Wallis tests were used as appropriate to compare both continuous and categorical variables.

All tests were two-tailed, with statistical significance set at p < 0.05. We conducted all analyses using MedCalc software for Windows, version 20.009 (MedCalc Software, Ostend, Belgium).

## Results

### Study population

**Clinical characteristics of patients.** The study population consisted of 306 patients. The most prevalent patient groups were acute leukemia (n = 117, 38.2%), HSCT (n = 71, 23.2%), and lymphoid malignancies (n = 52, 17.0%). Details on the patients' underlying conditions are available in S1 Table.

Clinical characteristics are displayed in Table 1. They varied between groups. The male/female ratio was 1.17:1, with 53.9% (165) men. However, there was a significantly higher percentage of women (84.6%, p = 0.001–0.04) in the solid cancer group. The median age was 55 years, with significant differences between groups (p<0.001): for instance, median age was 48 years in the HSCT group, and 65 in the MPN/MDS group (p<0.001).

**Table 1. Clinical and biological characteristics of the study population.**

| | Total cohort (n = 306) | Acute Leukemia group (n = 117) | HSCT group (n = 71) | Lymphoma, CLL, MM group (n = 52) | CAR T-cells group (n = 24) | MPN/ MDS group (n = 17) | Solid cancer group (n = 13) | Others group (n = 12) | p-value |
|---|---|---|---|---|---|---|---|---|---|
| **Male**, n (%) | 165 (53.9) | 61 (52.1) | 41 (57.8) | 32 (61.6) | 17 (70.8) | 9 (52.9) | 2 (15.4) | 3 (25.0) | 0.01 |
| Female, n (%) | 141 (46.1) | 56 (47.9) | 30 (42.2) | 30 (49.4) | 7 (29.2) | 8 (47.1) | 11 (84.6) | 9 (75%) | |
| **Age**, years | 55 (39.0–67.0) | 58 (45.8–69.3) | 48 (29.5–58.0) | 55.5 (39.5–69.0) | 61.5 (32.5–69.5) | 65 (57.0–69.0) | 56 (35.0–64.3) | 46 (35.0–70.0) | < 0.001 |
| **BMI**, kg/m² | 24.2 (22.0–27.6) | 24.2 (22.0–27.5) | 24.7 (22.0–27.6) | 24.5 (22.3–27.6) | 23.6 (22.0–27.0) | 25.0 (21.0–27.0) | 23.9 (21.2–28.2) | 23.1 (21.6–27.6) | > 0.99 |
| **Duration of neutropenia before CT***, days | 6 (2.0–12.0) | 9 (3.8–14.0) | 6 (3.0–10.0) | 3 (0.3–7.8) | 5.5 (2.0–9.0) | 12 (1.8–105.8) | 2 (0.8–6.5) | 4.5 (0.5–68.5) | 0.003 |
| **Prolonged neutropenia (> 14 days) before CT ****, n (%) | 53 (19.3) | 26 (24.5) | 10 (14.1) | 2 (5.0) | 2 (8.7) | 9 (52.9) | 1 (11.1) | 3 (37.5) | < 0.001 |
| **Duration of febrile neutropenia before CT†**, days | 2 (1.0–5.0) | 3 (1.0–5.0) | 2 (1.0–4.0) | 1.5 (0.0–5.0) | 2 (1.0–5.0) | 4 (1.0–7.0) | 1 (0.3–3.5) | 1 (0.0–8.0) | 0.52 |
| **WBC**, /mm³ | 340 (200–960) | 430 (200–1270) | 200 (200–330) | 320 (200–1040) | 450 (200–730) | 690 (370–1370) | 720 (290–1170) | 1040 (430–1930) | < 0.001 |
| **ANC ≤ 200/mm³ ††**, n (%) | 162 (52.9) | 58 (49.6) | 50 (70.4) | 26 (50.0) | 12 (50.0) | 6 (35.3) | 6 (46.2) | 4 (33.3) | 0.03 |
| **CRP level⁴**, mg/L | 113 (45.5–239.5) | 175 (69.0–265.0) | 100 (40.0–228.0) | 139 (45.0–286.0) | 90 (18.5–170.0) | 58 (21.0–151.8) | 124.5 (43.5–236.5) | 119.5 (76.0–203.5) | 0.14 |

HSCT: Hematopoietic stem cell transplantation, CLL: Chronic lymphoid leukemia, MPN: Myeloproliferative neoplasm, MDS: Myelodysplastic syndrome, BMI: Body mass index, WBC: White blood cell count, ANC: Absolute neutrophile count, CRP: C-reactive protein

Continuous variables are expressed as median (interquartile range). Percentages in each column are relative to the relevant disease group.

*Data available for 265 patients, **Data available for 274 patients, †Data available for 282 patients, ††Data available for 235 patients.

Overall, clinical signs (excluding fever) were reported in 180 (58.8%) patients, including nose/paranasal sinus signs in 9 (2.9%) patients, respiratory signs in 99 (32.4%), and digestive/urological signs in 105 (34.3%). Additionally, 237 (77.5%) patients were explored in at least one anatomical region in which they had no corresponding clinical sign (for instance explored with a chest CT without respiratory signs). One hundred and twenty-six patients (41.2%) had no clinical signs other than fever.

In total, 153 patients out of 306 finally received a clinical diagnosis of infection (50%), supported by radiological and/or microbiological evidence.

Changes in treatment following CT scans were reported in 84 (27.5%) patients.

**Biological characteristics of patients.** Biological characteristics of patients are detailed in Table 1. Data on neutropenia duration categorized as short or prolonged was available for 274 patients (89.5% of the total cohort), and 19.3% of patients (53/274) experienced prolonged FN (>14 days). The prevalence of prolonged FN was significantly higher in the MPN/MDS group (52.9% (9/17, p<0.001–0.02).

The median WBC count at the time of CT was 340/mm$^3$ (IQR 200–960). More than half of the patients (162/306, 52.9%) had an ANC $\leq$ 200/mm$^3$, with a notable overrepresentation in the HSCT group (50/71, 70.4%, p = 0.01–0.03).

## CT findings and relations to clinical signs, changes in treatment and final diagnoses

**CT characteristics.** Areas explored by CT included the chest region in 93.1% (285/306) of patients, the abdomen/pelvis region in 80.4% (246/306), and the paranasal sinus region in 22.2% (68/306). For 17.7% of patients (54/306), CT included the 3 regions. Intravenous contrast medium was used in 76.1% (233/306) patients, predominantly during examinations that included the abdominopelvic region (96.6%, 225/233).

**CT findings and clinical signs.** Overall, 188 patients (61.4%) had at least one positive CT finding, compatible with infection, on nose/paranasal sinus, chest and/or abdomen/pelvis regions. In patients with at least one clinical sign (in addition to fever), 126 out of 180 (66.7%) had at least one positive CT finding in the corresponding anatomical region (paranasal sinus CT finding for nose/paranasal sinus signs, chest CT finding for respiratory signs and abdomen/pelvis CT finding for digestive/urological signs). In patients without clinical signs in an explored anatomical region, 84 out of 237 (35.4%) had at least one positive CT finding in those regions. Finally, among the 126 patients without any clinical sign apart from fever, 51 (40.5%) had at least one positive CT finding.

The frequencies of CT positive findings for each anatomical region and the presence or absence of clinical signs in the corresponding system are detailed in Table 2. For the paranasal sinus region, 33.3% (3/9) of scans were positive in symptomatic patients versus 8.5% (5/59) in asymptomatic patients (p = 0.03). Chest and abdomen/pelvis CTs with positive findings were significantly more prevalent in patients with clinical signs compared with those without (65.7% vs. 25.8%, and 59.0% vs. 22.7% respectively, p<0.001). However, a notable percentage

**Table 2. CT abnormalities by presence of symptoms and by anatomical region.**

| Body region | Number of CT scans | No specific symptoms | | Presence of specific symptoms | | p-value |
|---|---|---|---|---|---|---|
| | | No CT abnormality | CT abnormality | No CT abnormality | CT abnormality | |
| **Paranasal sinus** | 68 | 54 (91.5%) | 5 (8.5%) | 6 (66.7%) | 3 (33.3%) | 0.03 |
| **Chest** | 285 | 138 (74.2%) | 48 (25.8%) | 34 (34.3%) | 65 (65.7%) | < 0.001 |
| **Abdomen/pelvis** | 246 | 109 (77.3%) | 32 (22.7%) | 43 (41.0%) | 62 (59.0%) | < 0.001 |

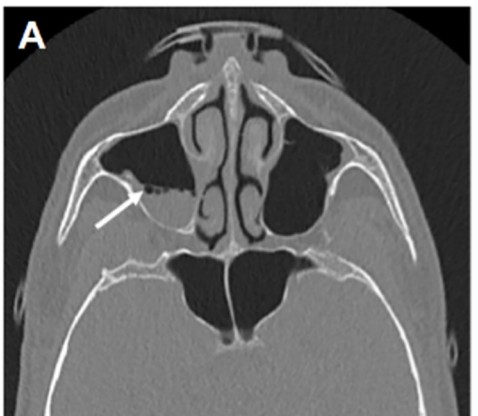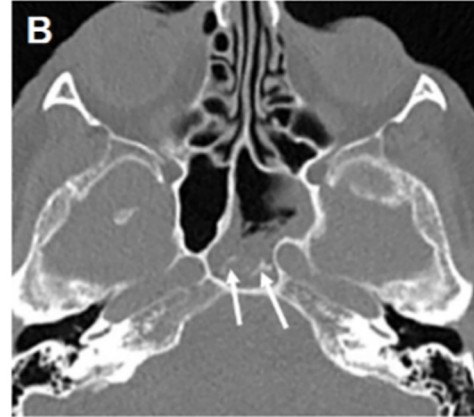

**Fig 2. Examples of CT findings in sinusitis cases.** Axial sections of paranasal sinus CT in 2 different patients (A, B). A: Partial filling of the right maxillary sinus with gas-fluid level and tiny bubbles on the surface of the fluid (arrow); final diagnosis of acute right maxillary sinusitis without microbiological evidence. B: Partial filling of the left portion of the sphenoidal sinus, with peripheral mucosal thickening, gas-fluid level, and some bubbles on the surface of the fluid; punctate hyper attenuations within the filling (arrow); final diagnosis of acute left sphenoidal sinusitis without microbiological evidence.

of positive CTs occurred in asymptomatic patients: 25.8% (48/186) for chest CTs and 22.7% (32/141) for abdomen/pelvis CTs. There was no significant difference in the distribution of positive CT findings among symptomatic and asymptomatic patients across the 7 patient groups.

**CT findings and final clinical diagnosis.** Among the 8 sinus CTs with positive findings, 5 resulted in a clinical diagnosis of acute sinusitis (3/5 in asymptomatic patients, 2/3 in symptomatic patients), without any proven case of invasive fungal infection. Illustrative examples are presented in Fig 2.

As shown in Table 3, 38 CT scans in the chest region (33.6%) out of 113 with positive findings did not lead to a clinical diagnosis of pneumonia. Among these, 25 (65.8%) were in patients without respiratory signs. Chest CT scans with positive findings in patients without respiratory symptoms (48/186, 25.8%) resulted in various diagnoses, including 4 cases (8.3%) of aspergillosis and 2 cases (4.2%) of mucormycosis.

Table 4 summarizes the causative agents found for the 75 pneumonias diagnosed in the cohort and Fig 3 illustrates the CT appearance of some of these infections.

Table 5 shows the semiology of CT abnormalities and corresponding final diagnoses in the abdomen/pelvis region. Twenty-eight CT scans (29.8%) out of 94 with positive CT findings did not lead to a diagnosis of infection in this region, mostly in patients with no clinical signs for the abdomen/pelvis region (20/28, 71.4%). Among patients with abdomen/pelvis positive CT findings, those related to a clinical diagnosis of neutropenic enterocolitis predominated, constituting 54.3% (51/94) of positive CT findings. There was one case each of hepatosplenic candidiasis and abdominal invasive mucormycosis (1.1% each). Fig 4 illustrates the CT appearance of some of these infections.

Finally, 39 patients out 118 (33.1%) were diagnosed with infection despite normal CTs, including bacteriemia or fungemia (catheter-related or not), musculo-skeletal, urinary and cutaneous infections. A summary of clinical signs, CT findings and final diagnoses is provided in S1 and S2 Figs.

**CT findings and treatment changes.** Table 6 presents the incidence of treatment changes following CT findings. Overall, treatment changes occurred in 84 instances of positive CT scan

**Table 3. Semiology of chest CT abnormalities and final diagnosis.**

| Semiology of CT abnormalities (n = 113) | Final diagnosis |
|---|---|
| **1. Nodules (n = 50)** | |
| a. Solid nodules (n = 15) | • Non-documented pneumonia (n = 4)<br>• *Aspergillus* fungal lung disease (n = 2)<br>• Viral lung disease (n = 1)<br>• Pulmonary tuberculosis (n = 1)<br>• Bacterial septic embolism (n = 1)<br>• No infectious diagnosis retained (n = 6) |
| b. Ground glass nodules (n = 4) | • Viral lung disease (n = 1)<br>• No infectious diagnosis retained (n = 3) |
| c. Solid nodules with halo sign (n = 17) | • *Aspergillus* fungal lung disease (n = 6)<br>• *Mucor* fungal lung disease (n = 2)<br>• Bacterial lung disease (n = 1)<br>• Bacterial septic embolism (n = 1)<br>• Non-documented pneumonia (n = 1)<br>• No infectious diagnosis retained (n = 6) |
| d. Cavitated nodules (n = 2) | • Non-documented pneumonia (n = 1)<br>• No infectious diagnosis retained (n = 1) |
| e. Focal micronodules (n = 12) | • Non-documented pneumonia (n = 4)<br>• Viral lung disease (n = 3)<br>• Viral lung disease with bacterial superinfection (n = 2)<br>• No infectious diagnosis retained (n = 3) |
| **2. Consolidations (n = 47)** | |
| a. Parenchymal consolidations (n = 36) | • Non-documented pneumonia (n = 14)<br>• Bacterial lung disease (n = 10)<br>• Viral lung disease (n = 3)<br>• Abscessed bacterial lung disease (n = 1)<br>• Septic embolism (n = 1)<br>• No infectious diagnosis retained (n = 7) |
| b. Consolidations and nodules (n = 7) | • Non-documented pneumonia (n = 2)<br>• Viral lung disease (n = 1)<br>• Viral lung disease with bacterial superinfection (n = 1)<br>• *Aspergillus* fungal lung disease (n = 2)<br>• No infectious diagnosis retained (n = 1) |
| c. Cavitated consolidations (n = 4) | • Non-documented abscessed pneumonia (n = 3)<br>• Abscessed bacterial lung disease (n = 1) |
| **3. Ground-glass opacities (n = 13)** | |
| | • Bacterial lung disease (n = 2)<br>• Non-documented pneumonia (n = 1)<br>• *Mucor* fungal lung disease (n = 1)<br>• *Aspergillus* fungal lung disease (n = 1)<br>• No infectious diagnosis retained (n = 8) |
| **4. Isolated pleural effusion (n = 3)** | • No infectious diagnosis retained (n = 3) |

**Table 4. Microbial etiology of diagnosed pneumonia.**

| | |
|---|---|
| **No microorganism isolated (40.0%, 30/75)** | • Non-abscessed pneumonia: 27 (36%)<br>• Abscessed pneumonia: 3 (4%) |
| **Bacterial lung disease (25.3%, 19/75)** | • Non-abscessed bacterial pneumonia: 13 (17.3%)<br>• Bacterial septic embolism: 3 (4%)<br>• Abscessed bacterial lung disease: 2 (2.7%)<br>• Pulmonary tuberculosis: 1 (1.3%) |
| **Fungal lung disease (18.7%, 14/75)** | *Aspergillus* pneumonia: 11 (14.7%)<br>• *Mucor* fungal lung disease: 3 (4%) |
| **Viral lung disease (16.0%, 12/75)** | • Viral lung disease: 9 (12%)<br>• Viral pneumonia with bacterial superinfection: 3 (4%) |

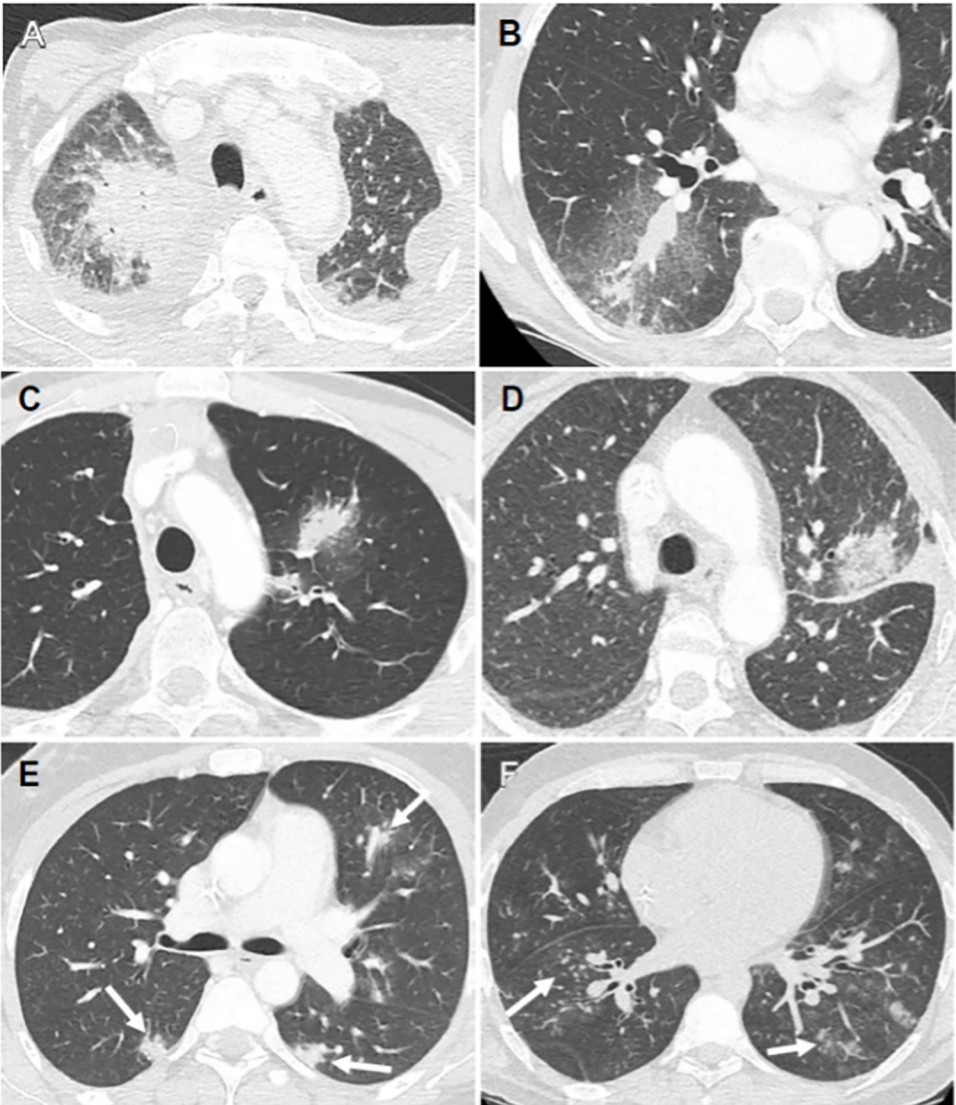

**Fig 3. Examples of CT findings in pneumonia cases.** Axial chest CT sections in 6 different patients (A-F). A: Parenchymal condensation of the right upper lobe, with irregular contours and aerated bronchogram, associated with bilateral pleural effusion. Microbiological diagnosis of *Pseudomonas aeruginosa* pneumonia. B: Parenchymal condensation of the right lower lobe with peripheral ground glass and intralobular reticulations (crazy paving). Microbiological diagnosis of *Staphylococcus warneri* pneumonia. C: Pseudonodular condensation area of the culmen, with peripheral ground glass halo. Microbiological diagnosis of *Aspergillus fumigatus* pneumonia. D: Rounded ground-glass area of the apicodorsal segment of the culmen, with irregular contours, associated with pleural effusion. Microbiological diagnosis of *Rhizomucor* and *Rhizopus* pneumonia. E: Bilateral nodules with blurred contours (arrows). Microbiological diagnosis of septic embolisms due to *Stenotrophomonas maltophilia*. F: Bilateral micronodules and ground glass nodules with blurred contours (arrows). Microbiological diagnosis of *Parainfluenza virus type 3* pneumonia.

regions out of 215 (39.1%). A statistically significant increase in changes was observed after positive CT findings in chest (41/113 or 36.3% versus 0/172 or 0.0%, p<0.001) and abdomen/pelvis (42/94 or 44.7% versus 0/152 or 0.0%, p<0.001) CTs. However, there was no significant difference when a paranasal sinus positive finding was found (1 positive CT finding out of 7, p = 0.12).

**Table 5. Semiology of abdomen/pelvis CT abnormalities and final diagnosis retained.**

| Semiology of CT abnormalities (n = 94) | Final diagnosis retained |
|---|---|
| **1. Digestive tract (n = 70)** | |
| a. Parietal thickening of small intestines and/or colon (n = 65) | • Neutropenic enterocolitis (n = 51)<br>• Clostridioides colitis (n = 3)<br>• No infectious diagnosis retained (n = 11) |
| b. Appendicular thickening with dilation (n = 3) | • Uncomplicated acute appendicitis (n = 2)<br>• Enterocolitis with reactive appendicitis (n = 1) |
| c. Parietal thickening of a colic diverticulum with fat stranding (n = 2) | • Acute diverticulitis (n = 1)<br>• Abscessed diverticulitis (n = 1) |
| **2. Liver, biliary tract, spleen (n = 12)** | |
| a. Splenic hypoenhancing areas (n = 4) | • No infectious diagnosis retained (n = 4) |
| b. Hepatic hypoenhancing nodules (n = 3) | • No infectious diagnosis retained (n = 3) |
| c. Multiple hepatic and/or splenic hypoenhancing nodules (n = 3) | • Hepatosplenic candidiasis (n = 1)<br>• Invasive mucormycosis (n = 1)<br>• No infectious diagnosis retained (n = 1) |
| d. Gallbladder wall thickening with dilation (n = 2) | • Uncomplicated acute cholecystitis (n = 1)<br>• No infectious diagnosis retained (n = 1) |
| **3. Kidneys and urinary tract (n = 8)** | |
| a. Renal hypoenhancing areas (n = 5) | • No infectious diagnosis retained (n = 5) |
| b. Hypoenhancing renal nodules (n = 1) | • Renal abscesses (n = 1) |
| c. Pyelic wall thickening (n = 2) | • Acute pyelonephritis (n = 1)<br>• No infectious diagnosis retained (n = 1) |
| **4. Others (n = 4)** | |
| a. Isolated peritoneal effusion (n = 2) | • No infectious diagnosis retained (n = 2) |
| b. Soft tissue collection: scrotum (n = 1), anal margin (n = 1) | • Perineal cellulitis and orchitis (n = 1)<br>• Anal margin abscess (n = 1) |

## Discussion

Our single-center retrospective study, including 306 patients undergoing CT scans during a FN episode, offers an overview of imaging findings and associated clinical diagnoses in an immunocompromised population. Overall, 188 patients (61.4%) had at least one positive CT finding. There were significantly more chest and abdomen/pelvis CT positive findings in symptomatic patients (65.7% vs. 25.8%, and 59.0% vs. 22.7% respectively, p <0.001), but also a significant number of positive CTs in asymptomatic patients (25.8% and 22.7% for chest and abdomen/pelvis, respectively). Treatment changes following chest and/or abdomen/pelvis CT were only noted in cases of positive CT findings, reinforcing the crucial role of imaging in the management of these patients.

Most patients in our study had hematological conditions, and solid cancer cases constituted only 4.2% of the cohort. The low representation of solid tumors is first due to an institutional factor, the center where the study was performed being an academic center with 23% of the 500 inpatient beds dedicated to hematology departments. Secondly, neutropenia in patients with solid tumors, usually attributable to chemotherapy, is less severe and less prolonged than in patients with hematological conditions receiving myeloablative treatments, with likely lower frequencies of CT use in their management [11].

Paranasal sinus CT scans (68, 22.2% of patients) were mostly performed in asymptomatic patients (54, 79.4%) and only 5 diagnoses of non-fungal acute sinusitis were made (3 in asymptomatic patients). Paranasal sinus involvement is not uncommon in invasive fungal infections [12] and mortality is high [13]. Paranasal sinus CT is therefore recommended by some

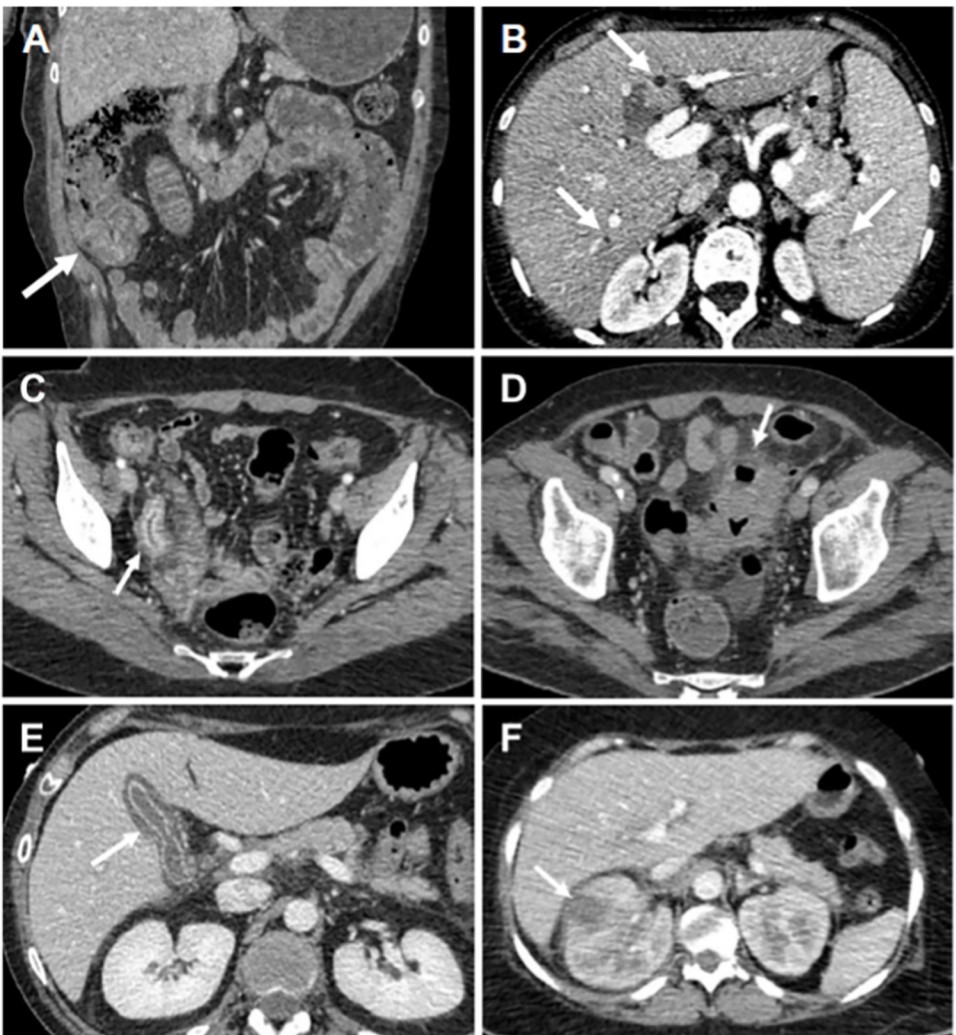

**Fig 4. Examples of CT findings in various abdominal diseases.** Abdomen/pelvis CTs in coronal (A) and axial (B-F) sections in 6 different patients. Abnormalities indicated by arrows. A: Wall thickening of the cecum: Diagnosis of neutropenic colitis. B: Newly appeared hypoenhancing hepatic and splenic micronodules: Diagnosis of hepatosplenic invasive candidiasis. C: Appendiceal dilation with wall thickening, mucosal hyperenhancement and submucosal edema: Diagnosis of acute appendicitis. D: Segmental wall thickening of the sigmoid colon with edematous thickening of a dilated diverticulum, and pericolic fat stranding: Diagnosis of uncomplicated acute diverticulitis. E: Gallbladder wall thickening, with mucosal hyperenhancement and perivesicular fat stranding: Diagnosis of acute cholecystitis. F: Hypo enhancing renal collection: Diagnosis of renal abscess.

**Table 6. Therapeutic changes undertaken after CT scans.**

| Region | | No change | Therapeutic modification | p-value |
|---|---|---|---|---|
| CT paranasal sinuses (n = 68) | No CT abnormality | 60 (88.2%) | 0 (0%) | 0.12 |
| | CT abnormality | 7 (10.3%) | 1 (1.5%) | |
| CT chest (n = 285) | No CT abnormality | 172 (60.4%) | 0 (0%) | <0.001 |
| | CT abnormality | 72 (25.3%) | 41 (14.4%) | |
| CT abdomen-pelvis (n = 246) | No CT abnormality | 152 (61.8%) | 0 (0%) | <0.001 |
| | CT abnormality | 52 (21.1%) | 42 (17.1%) | |

guidelines in case of neutropenia and evocative clinical signs. However, there is no guidance on the use of imaging in patients without localizing symptoms [14,15]. In this subgroup, our results, which underline the low diagnostic yield of paranasal sinus CT in asymptomatic patients, are in line with previously published data. In a study that included 56 paranasal sinus CT, Gurram et al found that the source of febrile neutropenia was attributed to the CT sinus findings in 9 cases, all of whom occurring in symptomatic patients [16]. Another work by Chang et al included 142 non targeted paranasal sinus CT in patients with febrile neutropenia and identified positive examinations in only 13 cases (9.1%) [17]. Moreover, it has been shown that sinusitis if much less likely to be a source of infection, compared to pneumonia [17,18]. Of note, in our study, none of the patients diagnosed with pulmonary or abdominal invasive fungal infection had concomitant positive paranasal sinus CT findings. To our knowledge, there is no relevant study evaluating the impact of paranasal sinus CT on morbidity and mortality in patients with FN, therefore its role in asymptomatic patients and in the absence of a prior diagnosis or suspicion of invasive fungal infection remains to be proven. Even in this case, it must be reminded that paranasal sinus CT has a suboptimal sensitivity and specificity for fungal infection at the early stage [19].

Lung is known as the first site of infection in immunocompromised patients [20] and chest was the most explored area (93.1%). Positive findings were found in 113 (39.6%) but approximately one-third of these abnormalities (38/113, 33.6%) did not translate into an infectious diagnosis, mostly in asymptomatic patients (25/38, 65.8% of this subgroup). Some chest abnormalities were not specific and could notably indicate features of disease specific lesions or of pulmonary edema, a common finding in patients with hematological malignancies [21]. It is also possible that microbiological sampling limitations did not allow to perform a diagnosis, particularly in case of minor abnormalities, and pneumonia diagnosis was therefore not retained [22]. Interestingly, 25.8% (48/186) of patients without respiratory signs had positive chest CTs, contributing to a clinical diagnosis of infection in nearly half of them (23/48, 45.8%), including 6 cases of fungal infections. In a recent article, Chang et al found an even higher 33.3% (46/138) rate of positive chest CT scans in patients with febrile neutropenia without localizing signs. The authors did not provide the number of fungal infections among this population [17]. However, in a cohort of patients recently diagnosed with acute myeloid leukemia, Bitterman et al. showed that the diagnosis of invasive pulmonary aspergillosis was not predicted by the presence or absence of respiratory symptoms. This underlines the poor predictive performances of clinical signs in this population. On the other hand, the role of early chest CT for the detection of pulmonary fungal infection is well established as a prompt diagnosis is critical to improve the disease outcome through the initiation of antifungal therapy [23,24]. Of note, it is now clear that chest X-ray has suboptimal performances for infection detection in this specific context [6,25,26]. Overall, current national and international guidelines recommend the use of chest CT in FN, with some variations: in cases of respiratory signs, or after 72h of appropriate antibiotics, or in patients at risk for invasive aspergillosis even without respiratory symptoms [2,14,15]. Our results certainly support the use of routine chest CT in the context of FN, including in asymptomatic patients. Chest CT is also valuable because of its excellent negative predictive value, allowing to rule out a lung infection as the cause of the fever. Indeed, in our cohort, we did not observe any change in the anti-infectious treatment when the CT showed no abnormality. Still, it should be repeated if this fever persists, as abnormalities related to pneumonia are likely to appear rapidly [27]. Low-dose protocols should be used to limit radiation exposure [28].

Finally, 246 abdomen/pelvis CTs were performed, i.e., in 80.4% of included patients, with 94 cases of positive findings (38.2%). As in chest CTs, there was a significantly higher frequency of positive findings in symptomatic cases compared with asymptomatic ones (59.0%

vs. 22.7%, p<0.001) and a substantial proportion of positive CT scans (28/94, 29.8%) that did not lead to an infectious diagnosis. It is essential to emphasize the nonspecific nature of CT abnormalities in leading to these diagnoses, underscoring the pivotal role of clinical signs and microbiological sampling. Interestingly, 22.7% (32/141) of abdomen/pelvis CT scans in asymptomatic patients were positive. The most common CT findings were intestinal abnormalities, as identified by previous studies that included patients with febrile neutropenia and CT of the abdominal region, with or without localizing clinical signs [17,29]. The most common clinical diagnosis associated with digestive wall thickening was neutropenic enterocolitis, identified in 54.3% (51/94) of abdomen/pelvis CT with positive findings. Neutropenic enterocolitis has previously been identified as the first cause of radiologic bowel abnormality in a cohort of neutropenic patients [30]. It is a severe complication of neutropenia, whose pathophysiology is still not fully understood, that favors infection; for this reason, broad-spectrum antibiotics are usually required, and antifungal therapy can be discussed in severe cases [10,31,32]. The diagnosis of neutropenic enterocolitis generally involves association of fever, abdominal pain, neutropenia, and thickening of the abdominal wall on abdominal studies (ultrasound and CT) [33–35]. Nevertheless, clinical signs may be absent, and imaging therefore plays a central role [36]. Abdominal ultrasound is a radiation-free, contrast-free method; however, it may be less performant than CT for the identification of complications that may require surgical intervention or for alternative diagnoses [10]. Aside from neutropenic enterocolitis, several cases of common acute abdominal infections were identified, including diverticulitis, appendicitis and cholecystitis. Such diagnoses may require specific interventions, beyond antimicrobial treatment. However, neutropenic patients may have unusual presentations, with attenuated clinical signs [37]. Finally, invasive fungal infections in the abdominopelvic region were uncommon in our cohort (2 cases). However, given their severity, early detection is important, and CT may have a role to play in it, in association with clinical and biological data, as well as in monitoring the response to treatment. Overall, and despite the lack of current recommendations on this topic, our study suggests that abdomen/pelvis CT may be useful in asymptomatic patients with febrile neutropenia.

The analysis of therapeutic modifications following CT scans revealed a significant association between the presence of abnormalities on chest and abdomen/pelvis CTs and a higher frequency of treatment changes in affected patients compared with those without abnormalities. We did not find any significant difference regarding paranasal sinus CT scans, possibly due to the smaller number of patients in that group. However, it is important to acknowledge that the absence of therapeutic modification does not necessarily indicate a lack of impact of imaging in patient management. In cases where radiological abnormalities with infectious features are identified, maintaining an appropriate antibiotic therapy may be prioritized over de-escalation. This also underscores the multifaceted role of CT findings in patient care. The radiological insights from CT scans can contribute to the prognostic evaluation of patients who already have a clinical diagnosis. Additionally, early detection of disease progression despite chemotherapy may prompt consideration of alternative therapies before scheduled oncologic reassessment. Still, caution is advised in interpreting the results related to post-CT therapeutic changes. The retrospective nature of clinical data, sourced solely from hospitalization reports, may not capture the nuances of complex therapeutic discussions, potentially oversimplifying the decision-making process and the impact of CT.

This study has several limitations, primarily due to its retrospective nature. Patient selection relied on indications mentioned in CT scan reports, and collection of clinical data, as stated above, may have been incomplete. Consequently, clinical signs that were present but not mentioned in clinical files can have been missed. Additionally, we could not control with precision the criteria on which clinical diagnoses of infection were made and had to rely on the

conclusion of the hospitalization reports. For this reason, it is also challenging to understand why some positive CT findings did not translate into an infection diagnosis. A second limitation pertains to the cohort study design, excluding FN patients who did not undergo CT scans. A control population would allow comparison of prognoses between those who underwent CT scans and those who did not. However, precise matching of all clinical and biological characteristics poses challenges, and ethical considerations also preclude such a study due to existing literature supporting the value of chest CT in this patient group. Finally, the single-center nature of the study further limits generalizability, with most patients having hematological malignancies. Applicability to patients with solid neoplasia remains uncertain, given the specificities of our hospital's practices and the lack of clear guidelines for non-chest CT scans in FN. Nonetheless, this study provides insights into the use of paranasal sinus and abdomen/pelvis CT scans in FN cases.

To conclude, our retrospective study showed that while chest and abdomen/pelvis CT abnormalities in patients with FN were more common in patients with clinical symptoms, a significant number of asymptomatic patients had positive findings in these regions that could lead to a clinical diagnosis of infection. These results support the existing recommendations for the large use of chest CT in patients with FN and may encourage the use of contrast-enhanced abdomen/pelvis CT in this indication, even in the absence of symptoms. Data regarding paranasal sinus CT, although more limited, do not provide a strong argument in favor of its systematic recommendation in patients without clinical signs. Because of the lack of specificity of some CT findings, communication between radiologists and referring physicians is key to interpret imaging data in challenging cases, to select the optimal management of these findings. However, questions remain unanswered. These questions include the actual impact of the imaging detection of abnormalities—particularly on abdomen/pelvis—on the prognosis of patients, as well as the cost-effectiveness of performing targeted versus non-targeted CT scans.

## Supporting information

**S1 Table. Patients' underlying conditions.**
(DOCX)

**S1 Fig. Summary of clinical signs, CT findings and final diagnoses, per region.**
(TIF)

**S2 Fig. Summary of clinical signs, CT findings and final diagnoses, per patient.**
(TIF)

## Author Contributions

**Conceptualization:** Éric de Kerviler, Anne Bergeron, Cédric de Bazelaire, Constance de Margerie-Mellon.

**Data curation:** Charles Tran.

**Formal analysis:** Charles Tran, Constance de Margerie-Mellon.

**Investigation:** Charles Tran, Constance de Margerie-Mellon.

**Methodology:** Constance de Margerie-Mellon.

**Supervision:** Éric de Kerviler.

**Writing – original draft:** Charles Tran, Constance de Margerie-Mellon.

**Writing – review & editing:** Éric de Kerviler, Anne Bergeron, Emmanuel Raffoux, Aliénor Xhaard, Cédric de Bazelaire, Constance de Margerie-Mellon.

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
