## [Decision Letter · Decision Letter 0]

13 Nov 2024

PONE-D-24-45229Contribution of paranasal sinus, chest, and abdomen/pelvis computed tomography in patients with febrile neutropeniaPLOS ONE

Dear Dr. de Margerie-Mellon,

Thank you for submitting your manuscript to PLOS ONE. After careful consideration, we feel that it has merit but does not fully meet PLOS ONE’s publication criteria as it currently stands. Therefore, we invite you to submit a revised version of the manuscript that addresses the points raised during the review process.

We look forward to receiving your revised manuscript.

Kind regards,

Cho-Hao Howard Lee, M.D.

Academic Editor

PLOS ONE

Reviewers' comments:

Reviewer's Responses to Questions

**Comments to the Author**

1. Is the manuscript technically sound, and do the data support the conclusions?

Reviewer #1: Yes

Reviewer #2: Partly

Reviewer #3: Yes

Reviewer #4: Yes

2. Has the statistical analysis been performed appropriately and rigorously? 

Reviewer #1: Yes

Reviewer #2: Yes

Reviewer #3: Yes

Reviewer #4: Yes

3. Have the authors made all data underlying the findings in their manuscript fully available?

Reviewer #1: Yes

Reviewer #2: No

Reviewer #3: Yes

Reviewer #4: Yes

4. Is the manuscript presented in an intelligible fashion and written in standard English?

Reviewer #1: Yes

Reviewer #2: Yes

Reviewer #3: Yes

Reviewer #4: Yes

5. Review Comments to the Author

Reviewer #1: This retrospective study assessed the diagnostic utility of CT imaging across three anatomical regions—paranasal sinuses, chest, and abdomen/pelvis—in 306 patients with febrile neutropenia at a single academic center. The key findings indicate that CT abnormalities were more frequent in symptomatic patients. However, a notable proportion of asymptomatic patients also exhibited positive findings that influenced clinical management, accounting for 25.8% in the chest and 22.7% in the abdomen/pelvis. This evidence supports the routine use of chest CTs in cases of febrile neutropenia and suggests a potential value for abdomen/pelvis CT scans even in the absence of localizing symptoms.

Modification Suggested

1. The study explicitly focuses on immunocompromised patients with febrile neutropenia, excluding those with normal immune function. It is recommended to clearly state in the abstract that the research is centered on the immunocompromised population.

2. The methodology section should describe how immunocompromised status was determined for the inclusion of participants.

These targeted revisions will enhance the manuscript’s scientific rigor and clinical applicability, maintaining the integrity of its core findings and conclusions.

Reviewer #2: Thank you for sharing this important work on the role of computed tomography (CT) in managing febrile neutropenia (FN) among immunocompromised patients. The study’s focus on imaging as a tool for infection detection in FN patients is timely and clinically valuable, particularly given the complexities of diagnosing infections in neutropenic states. I appreciate the detail and rigor in your methodology, as well as the relevance of your findings for guiding treatment decisions.

Merits

1. High Clinical Relevance: Your study addresses a critical gap in FN management, especially highlighting imaging as a potentially routine assessment tool. For clinicians, the insight into how specific CT findings may influence treatment decisions—particularly chest and abdomen/pelvis imaging—is invaluable.

2. Thorough Methodology: Including multiple anatomical regions and utilizing a structured retrospective analysis strengthens the evidence and reliability of your findings, making it more generalizable to clinical settings with similar patient populations.

3. Significant Practical Implications: By underscoring the influence of positive CT findings on treatment adjustments, you’re providing a basis for possibly expanding imaging practices in FN patients, which could help prevent missed infections or complications.

Recommendations for Major Revision

While this study is meaningful, there are areas where further detail and clarification could improve its impact and clarity for readers:

1. Provide a Clearer Justification for Paranasal Sinus Imaging: The inclusion of paranasal sinus CT in asymptomatic patients may need more justification. Expanding on any anticipated benefits and clarifying the clinical contexts where it could be useful would add depth and make the rationale clearer for practitioners.

2. Expand on Non-specific CT Findings: It would be helpful to discuss in greater detail the implications of non-specific CT findings, especially in cases without confirmed infection. This could guide clinicians on how to interpret incidental findings and weigh the risks of overtreatment or unnecessary interventions that may arise.

3. Discuss the Retrospective Design Limitations More Fully: The study’s retrospective nature is mentioned, but providing a more thorough analysis of how this design may impact results—particularly with regard to missing clinical notes or potential biases in treatment decisions—would enhance transparency and contextualize the findings.

4. Address the Systematic Imaging in Asymptomatic Patients: Your findings suggest that systematic imaging may be beneficial even in patients without clinical symptoms. Expanding on existing guidelines or providing a cost-effectiveness perspective on imaging in asymptomatic cases would strengthen the clinical relevance and utility of this recommendation.

5. Strengthen the Integration of Visual Data with the Text: The tables and figures in the manuscript are informative but could be better integrated within the text. Referring directly to these visuals within relevant sections could enhance readability and make the flow of information smoother for readers.

Recommendation: Major Revision

In sum, your study has strong clinical implications, and with additional clarification and context in certain areas, it could become a valuable reference for FN management. I appreciate your effort in conducting this research and encourage these revisions to enhance the manuscript’s clarity, accessibility, and relevance for a wide clinical audience. Thank you again for your hard work, and I look forward to seeing the revised version.

Reviewer #3: The study investigates the utility of computed tomography (CT) scans across three anatomical regions—paranasal sinus, chest, and abdomen/pelvis—in patients with febrile neutropenia (FN), examining their role in identifying infectious abnormalities and influencing treatment changes. The study addresses an important topic, given the high morbidity and mortality associated with FN and the challenges in identifying infectious sources in this population. The manuscript is well-structured and presents a thorough analysis of clinical, imaging, and therapeutic data in a large cohort. However, there are some areas where the manuscript could benefit from further clarification and refinement.

Minor Comments：

1. The authors advocate for routine use of chest and abdomen/pelvis CT in asymptomatic FN patients. While findings suggest that positive CT results are relatively common even in the absence of symptoms, it would be helpful to expand on the clinical relevance and potential cost-effectiveness of this approach. Additionally, providing context with more comparative data from the existing literature could strengthen this argument.

2. Although the authors acknowledge that the study does not evaluate prognosis directly, this is an important point that would benefit from further discussion. For instance, elaborating on how imaging findings might correlate with long-term outcomes or impact clinical decision-making could add depth to the conclusions.

3. The findings suggest that paranasal sinus CT had limited impact on treatment changes. The manuscript could benefit from a clearer discussion of whether this justifies excluding sinus CT in the absence of symptoms. Additionally, some comparison to guidelines or prior studies evaluating paranasal sinus imaging in FN patients would provide a stronger basis for these recommendations.

4. The retrospective nature of the study introduces certain limitations, such as the reliance on available clinical notes for determining symptoms and treatment changes. It would be useful to acknowledge specific confounding factors that may influence the interpretation of CT findings, especially regarding subjective decisions in the clinical diagnosis of infections.

Reviewer #4: This study evaluates the effectiveness of paranasal sinus, chest, and abdomen/pelvis CT scans performed during an episode of FN, in patients with or without specific clinical signs to assess potential infectious origin in these sites—their finding support for a systemic CT in these patients with or without symptoms.

This is a reasonable approach, however, as mentioned in the conclusion the potential side effects and high risk of CT which are not essential should be considered.

Several other concerns are noticed in this study:

1. In line 73, define other causes of immunosuppression and give all the examples you used in your criteria. The criteria need to be clear and avoid any misinterpretation

2. In line 74, define fever (temperature higher than?)

3. In line 78, please clarify why you only considered the first CT.

4. In line 104, please define the physician diagnostic criteria for infectious and non-infectious. What are the lab results and clinical findings in each group?

5. Line 126-130, when explaining the finding of abdominal CT, make sure to mention any corresponding diagnosis to each of these findings (e.g.: gallbladder thickening and distension correspond to cholecystitis)

6. In Table 1, make sure to include the female number % as it might lead to misjudgment of the table data.

7. Table 4 needs a title heading to be more comprehensive.

6. PLOS authors have the option to publish the peer review history of their article (what does this mean?). If published, this will include your full peer review and any attached files.

Reviewer #1: No

Reviewer #2: **Yes: **Xiaoyi Zhang, M.D.

Reviewer #3: No

Reviewer #4: No

---

## [Author Response · Author response to Decision Letter 0]

25 Nov 2024

Response to reviewers has been uploaded in a separate file, as requested.

---

## [Decision Letter · Decision Letter 1]

11 Dec 2024

Contribution of paranasal sinus, chest, and abdomen/pelvis computed tomography in patients with febrile neutropenia

PONE-D-24-45229R1

Dear Dr. Constance de Margerie-Mellon,

We’re pleased to inform you that your manuscript has been judged scientifically suitable for publication and will be formally accepted for publication once it meets all outstanding technical requirements.

Kind regards,

Cho-Hao Howard Lee, M.D.

Academic Editor

PLOS ONE

Reviewers' comments:

Reviewer's Responses to Questions

**Comments to the Author**

1. If the authors have adequately addressed your comments raised in a previous round of review and you feel that this manuscript is now acceptable for publication, you may indicate that here to bypass the “Comments to the Author” section, enter your conflict of interest statement in the “Confidential to Editor” section, and submit your "Accept" recommendation.

Reviewer #1: All comments have been addressed

Reviewer #2: (No Response)

Reviewer #3: All comments have been addressed

Reviewer #4: All comments have been addressed

2. Is the manuscript technically sound, and do the data support the conclusions?

Reviewer #1: Yes

Reviewer #2: Yes

Reviewer #3: Yes

Reviewer #4: Yes

3. Has the statistical analysis been performed appropriately and rigorously? 

Reviewer #1: Yes

Reviewer #2: Yes

Reviewer #3: Yes

Reviewer #4: Yes

4. Have the authors made all data underlying the findings in their manuscript fully available?

Reviewer #1: Yes

Reviewer #2: No

Reviewer #3: Yes

Reviewer #4: Yes

5. Is the manuscript presented in an intelligible fashion and written in standard English?

Reviewer #1: Yes

Reviewer #2: Yes

Reviewer #3: Yes

Reviewer #4: Yes

6. Review Comments to the Author

Reviewer #1: (No Response)

Reviewer #2: Merits

1. Clinical Relevance: This study addresses a critical aspect of managing FN, a common and potentially life-threatening complication in onco-hematology patients. By evaluating the diagnostic utility of different CT modalities, the study contributes to optimizing care pathways.

2. Comprehensive Analysis: The study systematically assesses the contribution of paranasal sinus, chest, and abdomen/pelvis CT scans to clinical decision-making, including treatment modifications, adding depth and rigor to the findings.

3. Well-Organized Presentation: The manuscript is structured logically, with clear descriptions of methods, results, and implications, making the data accessible and actionable for clinicians.

Suggestions for Minor Revisions

1. Clarify Patient Selection Criteria:

While the inclusion and exclusion criteria are well-detailed, the rationale for focusing on specific anatomical regions (paranasal sinus, chest, and abdomen/pelvis) could be expanded to provide more context.

2. Discuss Implications of Asymptomatic Findings:

The study highlights a significant proportion of positive CT findings in asymptomatic patients, particularly in the chest and abdomen/pelvis regions. Adding a discussion on the clinical management challenges these findings pose (e.g., balancing overdiagnosis versus missed infections) would enrich the discussion.

3. Expand on Cost-Effectiveness Considerations:

Given the widespread use of CT scans in FN, a brief discussion on the cost-effectiveness of systematic imaging for asymptomatic patients versus targeted imaging strategies could be valuable for policymakers and hospital administrators.

4. Address Radiation Exposure Risks:

The manuscript mentions the use of low-dose CT protocols but could briefly elaborate on strategies to minimize radiation exposure, particularly in populations requiring repeated imaging.

5. Enhance the Limitations Section:

While the limitations are well-addressed, including a discussion on potential biases introduced by the monocentric and retrospective design would provide a more balanced view. Additionally, the lack of a control group for patients who did not undergo CT scans could be acknowledged as a potential limitation.

Recommendation: Minor Revision

This manuscript offers a valuable contribution to understanding the role of CT imaging in FN management. With these minor revisions, it will be even more impactful for clinicians and researchers alike. Thank you for your hard work on this important topic, and I look forward to seeing the final version published.

Reviewer #3: The authors have adequately addressed all of my concerns. I support the publication of this manuscript. Best of luck!

Reviewer #4: (No Response)

7. PLOS authors have the option to publish the peer review history of their article (what does this mean?). If published, this will include your full peer review and any attached files.

Reviewer #1: No

Reviewer #2: **Yes: **Xiaoyi Zhang, MD

Reviewer #3: No

Reviewer #4: No

---

## [Editor Report · Acceptance letter]

20 Dec 2024

PONE-D-24-45229R1 

PLOS ONE

Dear Dr. de Margerie-Mellon, 

I'm pleased to inform you that your manuscript has been deemed suitable for publication in PLOS ONE. Congratulations! Your manuscript is now being handed over to our production team.

Kind regards, 

on behalf of

Dr. Cho-Hao Howard Lee 

Academic Editor

PLOS ONE